

# The lemur baseline: how lemurs compare to monkeys and apes in the Primate Cognition Test Battery

Claudia Fichtel[1,2,*], Klara Dinter[1,*] and Peter M. Kappeler[1,3]

[1] Behavioral Ecology & Sociobiology Unit, German Primate Center, Göttingen, Germany
[2] Leibniz-ScienceCampus Primate Cognition, Göttingen, Germany
[3] Department of Sociobiology/Anthropology, Johann-Friedrich-Blumenbach Institute of Zoology and Anthropology, Georg-August Universität, Göttingen, Germany
[*] These authors contributed equally to this work.

## ABSTRACT

Primates have relatively larger brains than other mammals even though brain tissue is energetically costly. Comparative studies of variation in cognitive skills allow testing of evolutionary hypotheses addressing socioecological factors driving the evolution of primate brain size. However, data on cognitive abilities for meaningful interspecific comparisons are only available for haplorhine primates (great apes, Old- and New World monkeys) although strepsirrhine primates (lemurs and lorises) serve as the best living models of ancestral primate cognitive skills, linking primates to other mammals. To begin filling this gap, we tested members of three lemur species (*Microcebus murinus, Varecia variegata, Lemur catta*) with the Primate Cognition Test Battery, a comprehensive set of experiments addressing physical and social cognitive skills that has previously been used in studies of haplorhines. We found no significant differences in cognitive performance among lemur species and, surprisingly, their average performance was not different from that of haplorhines in many aspects. Specifically, lemurs' overall performance was inferior in the physical domain but matched that of haplorhines in the social domain. These results question a clear-cut link between brain size and cognitive skills, suggesting a more domain-specific distribution of cognitive abilities in primates, and indicate more continuity in cognitive abilities across primate lineages than previously thought.

Corresponding author
Claudia Fichtel,
claudia.fichtel@gwdg.de

# INTRODUCTION

One central question in comparative cognition is why primates have evolved larger brains and superior cognitive skills compared to other equally-sized mammalian species (*Shettleworth, 2010*). Among primates, this effect is paralleled by a disproportionate increase in brain size from strepsirrhines to haplorhines and humans (*Dunbar, 1992*; *Isler et al., 2008*; *Jerison, 1973*; *Martin, 1981*). Because larger brains are energetically more expensive (*Aiello & Wheeler, 1995*), they are assumed to confer benefits with regard to enhanced cognitive abilities that compensate this additional investment (*Navarrete, Van Schaik & Isler, 2011*; *Reader & Lal, 2002*; *Reader, Hager & Lal, 2011*).

Several non-mutually exclusive hypotheses on the evolution of brain size have been proposed to account for the distinctive cognitive abilities of primates (*Dunbar & Shultz, 2017*). According to the *General intelligence hypothesis*, larger brains are thought to confer an advantage because of faster learning and larger memory capacities (*Spearman, 1904*). The *Ecological intelligence hypothesis* suggests that environmental and ecological challenges in food acquisition, including spatial and spatio-temporal processes to memorize seasonally available food or manipulative skills for extractive foraging, selected for larger brains (*Byrne, 1996*; *Clutton-Brock & Harvey, 1980*; *Heldstab et al., 2016*; *Milton, 1981*; *Powell, Isler & Barton, 2017*). Several versions of the *Social brain hypothesis* posit that increased cognitive skills in primates evolved in response to the constant challenges associated with the complexity of social life, such as competition and cooperation within larger social groups (*Byrne & Whiten, 1988*; *Dunbar, 1992*; *Dunbar & Shultz, 2007*; *Humphrey, 1976*; *Jolly, 1966a*; *Kudo & Dunbar, 2001*). However, support for the *Social brain hypothesis* is not uniform in other taxa, with brain size correlating positively with measures of sociality in some insectivores, bats and ungulates (e.g., *Barton, Purvis & Harvey, 1995*; *Byrne & Bates, 2010*; *Dunbar & Bever, 1998*; *Shultz & Dunbar, 2006*, but not in corvids (*Emery et al., 2007*; *Shultz & Dunbar, 2007*), and it is equivocal in carnivores (*Benson-Amram et al., 2016*; *Dunbar & Bever, 1998*; *Finarelli & Flynn, 2009*; *Holekamp, Sakai & Lundrigan, 2007*; *Pérez-Barbería, Shultz & Dunbar, 2007*). Moreover, recent comparative analyses among primates indicated that brain size is associated with ecological (home range size, diet, activity period), but not with social factors (*DeCasien, Williams & Higham, 2017*; *Powell, Isler & Barton, 2017*), also challenging the social brain hypothesis.

Since these studies usually link interspecific variation in brain size with certain socio-ecological factors, it is essential to understand how brain size actually impacts cognitive skills. Hence, comparative studies of cognitive abilities, ideally using identical tests, across the primate order and beyond are required. However, comparisons of performance in cognitive experiments across species may fail due to variation in the experimental set-up and specific methods (*Van Horik & Emery, 2011*; *Krasheninnikova et al., 2019*; *MacLean et al., 2009*; *Schubiger, Fichtel & Burkart, 2020*).

To overcome this problem, Herrmann and colleagues (*2007*) assembled a systematic toolbox for comparative analysis, called the *Primate Cognition Test Battery* (PCTB), which compared cognitive skills in various tasks in the physical and social domain among 2.5-year-old children, chimpanzees (*Pan troglodytes*) and orangutans (*Pongo pygmaeus*). The physical domain deals with the spatial–temporal-causal relations of inanimate objects, while the social domain deals with the intentional actions, perceptions, and knowledge of other animate beings (*Tomasello & Call, 1997*). These tests revealed that children and chimpanzees have similar cognitive skills for dealing with the physical world, but children have increased cognitive skills for dealing with the social world, particularly in the scale of social learning. These results support the *Cultural intelligence hypothesis*, a variant of the Social brain hypothesis, suggesting that exchanging knowledge within human cultural groups requires specific socio-cognitive skills, such as social learning or Theory of Mind (e.g., *Boyd & Richerson, 1988*; *Herrmann et al., 2007*; *Whiten & Van Schaik, 2007*). The PCTB has been replicated in another population of chimpanzees, and this study also

indicated that variation in cognitive performance is heritable (*Hopkins, Russell & Schaeffer, 2014*).

Application of the PCTB to two other haplorhine species, long-tailed macaques (*Macaca fascicularis*) and olive baboons (*Papio anubis*), revealed that both species performed similarly to great apes in both the physical and the social domain (*Schmitt, Pankau & Fischer, 2012*). Specifically, chimpanzees outperformed macaques only in tasks on spatial understanding and tool use. Since chimpanzees have relatively larger brains than macaques or baboons (*Isler et al., 2008*; *Jerison, 1973*), these results question the clear-cut relationship between cognitive performance and brain size (*Schmitt, Pankau & Fischer, 2012*). In addition, four closely related macaque species that differ in their degree of social tolerance, performed similarly in tests of the PCTB in the physical domain. However, socially more tolerant species performed better in one task of the social domain and the inhibitory control task, suggesting that social tolerance is associated with a set of cognitive skills that are specifically required for cooperation (*Joly et al., 2017*). Thus, further studies on additional non-human primates are required to explore the interrelationships among cognitive abilities, socio-ecological traits and brain size (*ManyPrimates et al., 2019a*; *ManyPrimates et al., 2019b*).

Strepsirrhine primates are the obvious candidates for such an extended comparative approach because they represent the best living models of the earliest primates and the link between primates and other mammalian orders (*Fichtel & Kappeler, 2010*; *MacLean, Merritt & Brannon, 2008*). Strepsirrhines split off from the main primate lineage approximately 60 million years ago and retained many ancestral primate traits (*Martin, 1990*; *Yoder et al., 1996*; *Yoder & Yang, 2004*). Importantly, strepsirrhine primates have relatively smaller brains than haplorhines, and their brain size does not correlate with group size (*MacLean et al., 2009*). Although older studies suggested that strepsirrhine primates possess physical cognitive abilities that are inferior to those of haplorhines (e.g., *Ehrlich, Fobes & King, 1976*; *Jolly, 1964*; *Maslow & Harlow, 1932*), recent studies indicated that their cognitive skills are similar to those of haplorhines (e.g., *Deppe, Wright & Szelistowski, 2009*; *Fichtel & Kappeler, 2010*; *Kittler, Schnoell & Fichtel, 2015*; *Kittler, Kappeler & Fichtel, 2018*; *Santos, Barnes & Mahajan, 2005a*; *Santos, Mahajan & Barnes, 2005b*). However, existing studies of strepsirrhine cognition used isolated tests, hampering systematic interspecific comparisons. Hence, a comprehensive study investigating a broad variety of tasks addressing different cognitive skills in lemurs, and replicating the exact same methods used in the PCTB, seems warranted for a systematic comparison across both primate suborders.

To this end, we applied the PCTB to three species of lemur that differ in key socio-ecological traits: ring-tailed lemurs (*Lemur catta*), black-and-white ruffed lemurs (*Varecia variegata*; hereafter: ruffed lemurs) and gray mouse lemurs (*Microcebus murinus*, Table 1). Mouse lemurs have one of the smallest brains among primates, and absolute brain size increases from mouse lemurs over ring-tailed lemurs to ruffed lemurs (*Isler et al., 2008*). Ring-tailed lemurs are diurnal opportunistic omnivores that live in groups of on average 14 individuals (*Gould, Sussman & Sauther, 2003*; *Jolly, 1966b*; *Sussmann, 1991*). Ruffed lemurs are diurnal, frugivorous and live in groups (average 24 individuals), exhibiting a fission–fusion social organization (*Baden, Webster & Kamilar, 2015*; *Holmes et al., 2016*;
**Table 1** Summary of the most important socio-ecological traits of the seven non-human primate species discussed in the present study.

| Species | N (present study) | ECV (cc) | % fruit | dietary breadth | social system | average group size |
|---|---|---|---|---|---|---|
| chimpanzees (*Pan troglodytes*) | 106 | 368.4 | 66 | 6 | group | 47.6 |
| orangutans (*Pongo pygmaeus*) | 32 | 377.4 | 64 | 6 | solitary | 1.5 |
| olive baboons (*Papio anubis*) | 5 | 167.4 | 62 | 6 | group | 69 |
| long-tailed macaques (*Macaca fascicularis*) | 10-13 | 64 | 66.9 | 5 | group | 26 |
| ruffed lemurs (*Varecia variegata*) | 13 | 32.1 | 92 | 4 | group | 24 |
| ring-tailed lemurs (*Lemur catta*) | 26-27 | 22.9 | 54 | 5 | group | 14 |
| grey mouse lemurs (*Microcebus murinus*) | 9-16 | 1.6 | 31.3 | 4 | solitary | 1 |

**Notes.**
n, number of individuals; ECV, endocranial volume (absolute brain size); % fruit, percentage of fruit in the diet.
Data from: *Baden, Webster & Kamilar, 2015*; *Dammhan & Kappeler, 2008*; *Baden, Webster & Kamilar, 2015*; *Isler et al., 2008*; *Lahann, 2007*; *MacLean et al., 2013*; *Radespiel et al., 2006*; *Schmitt, Pankau & Fischer, 2012*.

*Vasey, 2003*). Gray mouse lemurs are nocturnal, omnivorous solitary foragers that form sleeping-groups composed of related females (*Eberle & Kappeler, 2006*).

According to the *General intelligence hypothesis*, we predicted that the tested apes and monkeys outperform lemurs because they have absolutely larger brains (Table 1). In accordance with the *Ecological intelligence hypothesis* we predicted that the more frugivorous species or those with a broader dietary breadth perform better (Table 1). Because lemurs generally live in smaller groups than monkeys and apes (*Kappeler & Heymann, 1996*), we predicted that they should have inferior cognitive abilities than the already tested group-living species according to the *Social intelligence hypothesis* (Table 1).

## METHODS

Experiments were conducted with adult individuals of gray mouse lemurs ($n = 9$–15), ring-tailed lemurs ($n = 26$–27) and black-and-white ruffed lemurs ($n = 13$). All individuals were born in captivity and housed in enriched or semi-natural environments, either at the German Primate Centre (DPZ, Göttingen) or the Affenwald Wildlife Park (Straußberg, Germany). The lemurs at the Affenwald range freely within a 3.5 ha natural forest enclosure. At the DPZ, ring-tailed and ruffed lemurs are offered indoor and outdoor enclosures equipped with enriching climbing materials and natural vegetation. The nocturnal mouse lemurs are kept indoors with an artificially reversed day-night-cycle, and their cages are equipped with climbing material, fresh natural branches and leaves. All individuals were tested individually in their familiar indoor enclosures and were trained to indicate their choice by touching or reaching for the chosen object, but were naïve to the presented tasks. Since some individuals passed away during the course of the study, not all individuals participated in every task of the test battery (Table S1, Supplemental).

To ensure comparability with the previous studies, the experimental setup was replicated after the PCTB (*Herrmann et al., 2007*; *Schmitt, Pankau & Fischer, 2012*), and only objects presented in the tests were adjusted in size for lemurs.

## Ethical statement

All animal work followed relevant national and international guidelines. The animals were kept under conditions documented in the European Directive 2010/63/EU (directive on the protection of animals used for experimental and other scientific purposes) and the EU Recommendations 2007/526/EG (guidelines for the accommodations and care of animals used for experimental and other scientific purposes). Consultation and approval of the experimental protocols by the Animal Welfare Body of the German Primate Center is documented (E2-17).

## General testing procedure

During the experiments, individuals were briefly separated from the group. The testing apparatus for all tasks consisted of a table with a sliding board on top that was attached to the mesh of the subjects' enclosures (Fig. S2). In most of the tasks two or three opaque cups (ruffed- & ring-tailed lemurs: Ø 6.8 cm × 7.5 cm; mouse lemurs: Ø 2.5 cm × 3 cm), which were placed upside down in a row on the sliding board, were used to cover the food reward (see also Supplementary Information). If necessary, a cardboard occluder was put on top of the sliding board between the experimental setup and the individual to hide the baiting process from the individuals. The position of the reward was randomized and counter-balanced across all possible locations, and the reward was never put in the same place for more than two consecutive trials. Once the board was pushed into reach of an individual, the experiment began, and, depending on the task, the individual had to manipulate an item or indicate its choice by pointing or reaching towards the chosen item, to obtain the reward if chosen correctly. If the choice was incorrect, the correct location of the reward was shown to the individual after each trial.

For most of the tasks at least 6 trials were conducted per individual and setup (Table S1). Raisins and pieces of banana served as rewards. During testing, no possible cues to where the reward was located were provided by the experimenter; she simply put her hands on her lap and her gaze was directed downwards. All experiments were videotaped and responses of the subjects to the tasks coded afterwards from the videos. A naïve second observer additionally scored 20% of all trials a second time to assess inter-observer reliability. The Interclass Correlation Coefficient was excellent (ICC = 0.985).

## The Primate Cognition Test Battery

All experimental setups and methods were replicated from the *PCTB* (*Herrmann et al., 2007*; *Schmitt, Pankau & Fischer, 2012*). Following *Schmitt, Pankau & Fischer (2012)*, we also doubled the number of trials for all object-choice tasks of the test battery (Table S1) to evenly distribute objects between all possible spatial positions and combinations of manipulations. In total, the PCTB consists of 16 different experimental tasks, 10 investigating physical and 6 social cognitive skills. These tasks can be grouped into

6 different scales: space, quantities and causality for the physical and social learning, communication and Theory of Mind for the social domain.

In the *physical domain*, the *space scale* examines the ability to track objects in space in four tasks: spatial memory, object permanence, rotation and transposition. The *quantities scale* tests the numerical understanding of individuals and consists of two tasks: relative numbers and addition numbers. The *causality scale* consists of four tasks: noise, shape, tool use and tool properties to examine the ability to understand spatial-causal relationships. In the *social domain*, the *social learning scale* examines in one task whether individuals use social information provided by a human demonstrator to solve a problem. The *communication scale* examines whether individuals are able to understand communicative cues given by humans in three tasks: comprehension, pointing cups and attentional state. Finally, in the *Theory of Mind scale*, individuals were confronted with two tasks: gaze following and intentions. A detailed description of the general setup and the methodology of the experiments can be found in (Supplementary Information).

## Temperament, inhibitory control, rank and learning effect

To assess the influence of temperament, inhibitory control and dominance rank on lemurs' performances in the test battery, individuals participated in a set of additional tests (*Herrmann et al., 2007*; *Schmitt, Pankau & Fischer, 2012*). Due to logistic constraints, the temperament tests could only be conducted with ring-tailed and ruffed lemurs. For temperament, we measured whether individuals would approach novel objects, people and food (for details see Supplementary Information). Inhibitory control was measured during an additional session of the spatial memory task, in which out of three cups only the two outer ones were baited with a reward, and, hence, individuals had to skip the cup in the middle. Dominance rank (high, middle or low-ranking) was inferred by focal observations of ring-tailed and ruffed lemurs but not for the solitary mouse lemurs, according to *Pereira & Kappeler (1997)*. We also controlled for potential learning effects within the trials of a task by calculating Pearson's correlations between performance in the first and second half of trials.

## Data analyses

We measured the performance of individuals by the proportion of correct responses for each task. We applied Wilcoxon tests followed by Benjamini–Hochberg corrections (for multiple testing) for each task and lemur species to examine whether they performed above chance level. Since no individual solved the social learning task and only one the tool use task, we omitted both tasks from the interspecies comparisons. To analyse whether the three lemur species differed in their performance in the tasks of the PCTB, we used multivariate analysis of variance (MANOVA) with species, sex, rank, age and age:species as between-subject factor and their performance in all tasks as dependent variable. To compare all three species' performances between the different tasks, we used univariate analysis of variance (ANOVA, for normally distributed data) or Kruskall–Wallis tests followed by *post hoc* analyses (with Bonferroni correction). For significant results, we used an analysis of covariance (ANCOVA) to control for age in these tasks.

Comparisons of performance in tests of the PCTB were conducted between the three lemur species and four haplorhine species (chimpanzees, orangutans, olive baboons, and long-tailed macaques) for which data on individual performance were kindly provided by E. Herrmann and V. Schmitt. On the scale level, we applied a MANOVA, followed by ANOVAs or Kruskall-Wallis tests and *post hoc* corrections (Bonferroni) in case of significant results. All statistical analyses were conducted in R version 3.2.2 (R Core Team, Vienna, Austria).

## RESULTS

### Lemurs' performance in the physical and social domain

In the *physical domain*, the chance level was at 33% in all four tasks of the *space scale*. The three lemur species performed significantly above chance level in the spatial memory and the rotation task (Table 2, Fig. 1). In the object permanence tasks, only ruffed lemurs performed above chance level, while in the control task, all three species performed above chance level (Table 2, Fig. 1). In the *quantities scale*, the three lemur species performed significantly above chance level (50%) in both tasks (Table 2, Fig. 1). In the *causality scale*, the tool use task was successfully solved by only one ring-tailed lemur. However, in the shape and tool properties tasks, all three lemur species performed above chance level (50%; Table 2).

In the *social domain*, no lemur solved the social learning task using a similar technique as demonstrated by a human experimenter (Table 2, Fig. 1). In the *communication scale*, all three lemur species performed significantly above chance level (50%) in the comprehension task, whereas only mouse lemurs performed above chance level (50%) in the pointing cups task. All lemur species performed poorly in the attentional state task. In the *Theory of Mind scale,* none of the lemur species did follow the gaze of the human experimenter upwards significantly more often than in the control condition in which no cue was given (baseline: 20%; Table 2, Fig. 1). In contrast, all lemur species performed significantly above chance level (50%) in the intentions task (Table 2, Fig. 1).

### Influence of age, sex and rank on performance of the three lemur species

Because the tool use task was solved by only one individual and the social learning task by none, these two tasks were excluded from this comparison. A multivariate analysis of variance of the 14 remaining tasks revealed no differences in the average performance among the three lemur species (MANOVA; Wilk's $\Lambda = 0.498$, $F (19,14) = 1.37$, $p = 0.257$). Furthermore, average performance was not influenced by sex (Wilk's $\Lambda = 0.461$, $F (19,14) = 1.59$, $p = 0.173$), rank (Wilk's $\Lambda = 0.273$, $F (38,28) = 1.24$, $p = 0.268$), age (Wilk's $\Lambda = 0.568$, $F (19,14) = 1.03$, $p = 0.466$) or age within species (age:species; Wilk's $\Lambda = 0.599$, $F (19,14) = 0.91$, $p = 0.566$).

### Personality, inhibitory control and learning

The three temperament measures (latency, proximity and duration) of ring-tailed or ruffed lemurs did neither correlate with the performance in the physical domain of the

Fichtel et al. (2020), *PeerJ*, DOI 10.7717/peerj.10025

Peer J

**Table 2** Summary of the mean proportions of correct responses of the three lemur species in all tasks and *scales* of the PCTB.

| | Trials | Chance | Ruffed lemurs | | | | | Ring-tailed lemurs | | | | | Mouse lemurs | | | | |
|---|---|---|---|---|---|---|---|---|---|---|---|---|---|---|---|---|---|
| | | | n | M | adj p | SD | 95% CI | n | M | adj p | SD | 95% CI | n | M | adj p | SD | 95% CI |
| Physical domain | | | | | | | | | | | | | | | | | |
| *Space* | | | | 46.8 | | 8 | 51, 58 | | 44.2 | | 7 | 42. 47 | | 50.8 | | 7 | 47, 55 |
| Spatial memory | 6 | 33 | 13 | **53.9** | 0.017 | 23 | 42, 66 | 27 | **55.6** | 0.001 | 17 | 49, 62 | 15 | **66.7** | 0.004 | 18 | 58, 68 |
| Object permanence | 18 | 33 | 13 | **47.9** | 0.006 | 12 | 41, 55 | 27 | 38.3 | 0.112 | 15 | 32, 44 | 12 | 42.1 | 0.074 | 10 | 36, 48 |
| Rotation | 18 | 33 | 13 | **45.3** | 0.014 | 10 | 40, 51 | 26 | **41.0** | 0.002 | 9 | 37, 45 | 12 | **47.7** | 0.008 | 9 | 43, 53 |
| Transposition | 18 | 33 | 13 | 40.2 | 0.052 | 13 | 33, 47 | 27 | **42.2** | 0.001 | 11 | 38, 46 | 12 | **41.2** | 0.019 | 12 | 35, 48 |
| *Quantities* | | | | 66.4 | | 12 | 60, 73 | | 58.5 | | 11 | 54, 63 | | 63.9 | | 6 | 60, 68 |
| Relative numbers | 16 | 50 | 13 | **62.0** | 0.006 | 7 | 58, 66 | 27 | **60.4** | 0.007 | 10 | 57, 64 | 9 | **66.0** | 0.019 | 11 | 59, 73 |
| Addition numbers | 14 | 50 | 13 | **70.9** | 0.014 | 20 | 60, 82 | 26 | **60.2** | 0.003 | 13 | 55, 65 | 9 | **61.9** | 0.019 | 8 | 57, 67 |
| *Causality* | | | | 51.0 | | 7 | 47, 55 | | 48.6 | | 7 | 46, 51 | | 44.0 | | 4 | 42, 46 |
| Noise | 12 | 50 | 13 | **63.5** | 0.015 | 13 | 56, 71 | 27 | **59.3** | 0.002 | 10 | 55, 63 | 15 | 50.0 | 0.958 | 17 | 41, 59 |
| Shape | 12 | 50 | 13 | **76.9** | 0.006 | 15 | 69, 85 | 27 | **72.8** | 0.001 | 10 | 69, 77 | 15 | **70.6** | 0.004 | 12 | 65, 77 |
| Tool use | 1 | – | 13 | 0.0 | - | - | - | 27 | 3.7 | – | 19 | −4, 11 | 15 | 0.0 | – | – | - |
| Tool properties | 30 | 50 | 13 | **63.6** | 0.013 | 12 | 57, 70 | 27 | **58.6** | 0.001 | 8 | 56, 62 | 15 | **55.6** | 0.040 | 9 | 51, 60 |
| Social domain | | | | | | | | | | | | | | | | | |
| *Social learning* | 3 | - | 13 | 0.0 | - | - | - | 26 | 0.0 | | - | - | 12 | 0.0 | | - | - |
| *Communication* | | | | 53.1 | | 12 | 47, 60 | | 49.6 | | 11 | 46, 54 | | 52.1 | | 9 | 47, 57 |
| Comprehension | 18 | 50 | 13 | **70.9** | 0.006 | 10 | 66, 76 | 27 | **70.8** | 0.001 | 13 | 66, 76 | 13 | **65.4** | 0.008 | 11 | 59, 72 |
| Pointing cups | 8 | 50 | 13 | 53.9 | 0.220 | 9 | 49, 59 | 27 | 55.1 | 0.050 | 12 | 51, 59 | 15 | **68.3** | 0.008 | 16 | 60, 76 |
| Attentional state | 4 | – | 13 | 34.6 | - | 28 | 19, 50 | 26 | 21.2 | – | 22 | 13, 30 | 14 | 25.0 | – | 22 | 14, 36 |
| *Theory of mind* | | | | 43.7 | | 10 | 45, 57 | | 56.8 | | 18 | 50, 64 | | 51.4 | | 11 | 39, 49 |
| Gaze following | 9 | 20 (bl) | 13 | 23.9 | 0.326 | 17 | 15, 33 | 27 | 30.0 | 0.340 | 33 | 18, 42 | 15 | 11.1 | 0.713 | 17 | 2, 20 |
| Intentions | 12 | 50 | 13 | **78.9** | 0.006 | 13 | 72, 86 | 27 | **83.6** | 0.001 | 15 | 78, 89 | 15 | **71.1** | 0.004 | 10 | 66, 76 |

**Notes.**

Numbers in boldface: Significant deviations from chance level (Wilcoxon tests).

Trials, number of trials per task; chance, chance-level for each task; n, number of participating individuals; M, means of performance; adj, adjusted *p*-values (Benjamini-Hochberg-corrections); SD, standard deviation; CI, confidence interval; bl, baseline calculated from control condition.

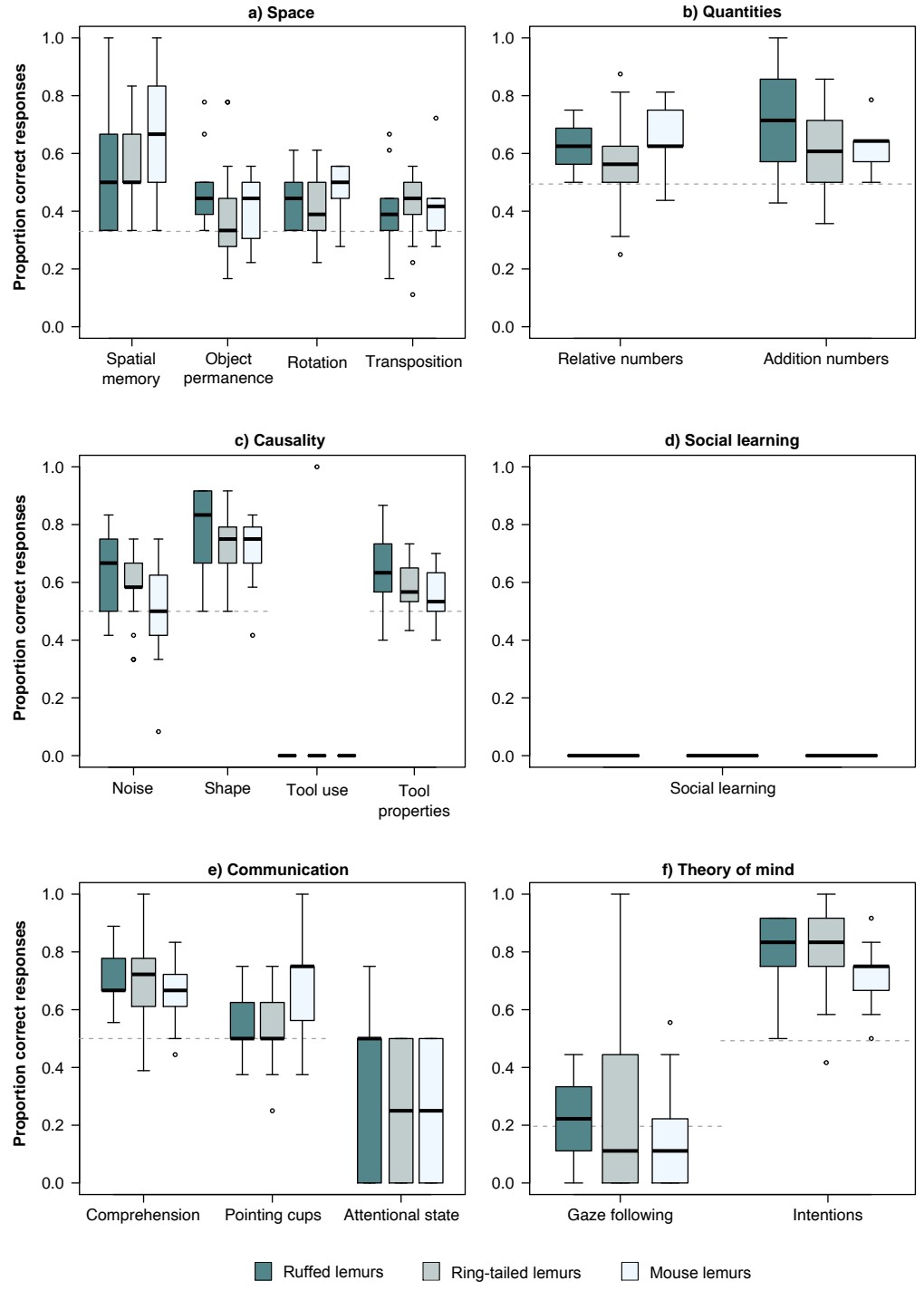

**Figure 1** **Performance in the PCTB of the three lemur species.** Average performance of the three lemur species in all tasks of the PCTB in the scales (A) Space, (B) Quantities, (C) Causality, (D) Social learning, (E) Communication, and (F) Theory of mind. Represented are medians (black bars), interquartile ranges (boxes), and outliers (circles).

PCTB (Pearson's correlations, all $p > 0.05$, see Supplementary Information), nor with the performance of ring-tailed lemurs in the *social domain*. In ruffed lemurs, however, the latency to approach and proximity to a novel stimulus correlated with performance in the social domain (latency to approach: Pearson's correlation, $r (11) = 0.61$, $p = 0.026$; proximity: Pearson's correlation, $r (11) = -0.59$, $p = 0.032$). No correlation was found between the time individuals spent close to the setup (duration) and performance in the social domain (Pearson's correlation, $r (11) = -0.30$, $p = 0.323$). Performance in the inhibitory control task did not correlate with performance in the physical and social domain (see Table S4). In addition, we did not find a learning effect in performance between the first and second half of trials within the tasks (Wilcoxon Signed-Rank test: $V = 806.5$, $p = 0.585$).

## Comparison of lemurs and haplorhines in the physical and social domain

The comparison of chimpanzees, orangutans, baboons, macaques, ruffed-, ring-tailed- and mouse lemurs in their overall average performance in the two domains revealed differences among species (Wilk's $\Lambda = 0.383$, $F (406,12) = 20.87$, $p < 0.001$). Species differed in performance in the *physical domain* (Kruskal–Wallis, $\chi^2 = 127.26$, $df = 6$, $p < 0.001$; Fig. 2), but not in the *social domain* (Kruskal–Wallis, $\chi^2 = 10.25$, $df = 6$, $p = 0.115$; Fig. 2). In the *physical domain*, only chimpanzees performed significantly better than ruffed lemurs, and chimpanzees and orangutans outperformed ring-tailed and mouse lemurs (see Table S4).

## Comparison of lemurs and haplorhines in the different scales

For a more detailed comparison of all seven species, we conducted a MANOVA including each individuals' overall performance in all six scales, which revealed significant differences among species (Wilk's $\Lambda = 0.284$, $F (833,36) = 7.68$, $p < 0.001$). Species differed in all scales except the *communication scale* (ANOVAs or Kruskal–Wallis tests, see Table 3; Fig. 3). In the *space scale*, chimpanzees outperformed all other species, except baboons. Orangutans performed better than ruffed and ring-tailed lemurs, baboons performed better than all three lemur species, and macaques performed similar to all lemur species (Table 4; Fig. 3). In the *quantities scale* , only chimpanzees performed better than ring-tailed lemurs (Table 4; Fig. 3), and in the *causality scale*, chimpanzees outperformed all other species, and orangutans performed better than mouse lemurs (Table 4; Fig. 3). However, this scale was strongly biased by the results of the tool use task, which was only solved by chimpanzees, orangutans and one ring-tailed lemur. Excluding the tool use task from this comparison revealed that only chimpanzees performed better than mouse lemurs (Table 4; Fig. S2).

In the *social domain,* all species, except great apes, performed poorly in the social learning task, whereas all species performed equally well in the *communication scale* (Table 4; Fig. 3). In the *Theory of Mind scale*, however, chimpanzees performed less well than macaques and ring-tailed lemurs. All other species performed better than orangutans, except mouse lemurs and macaques, and ring-tailed lemurs outperformed mouse lemurs (Table 4; Fig. 3).

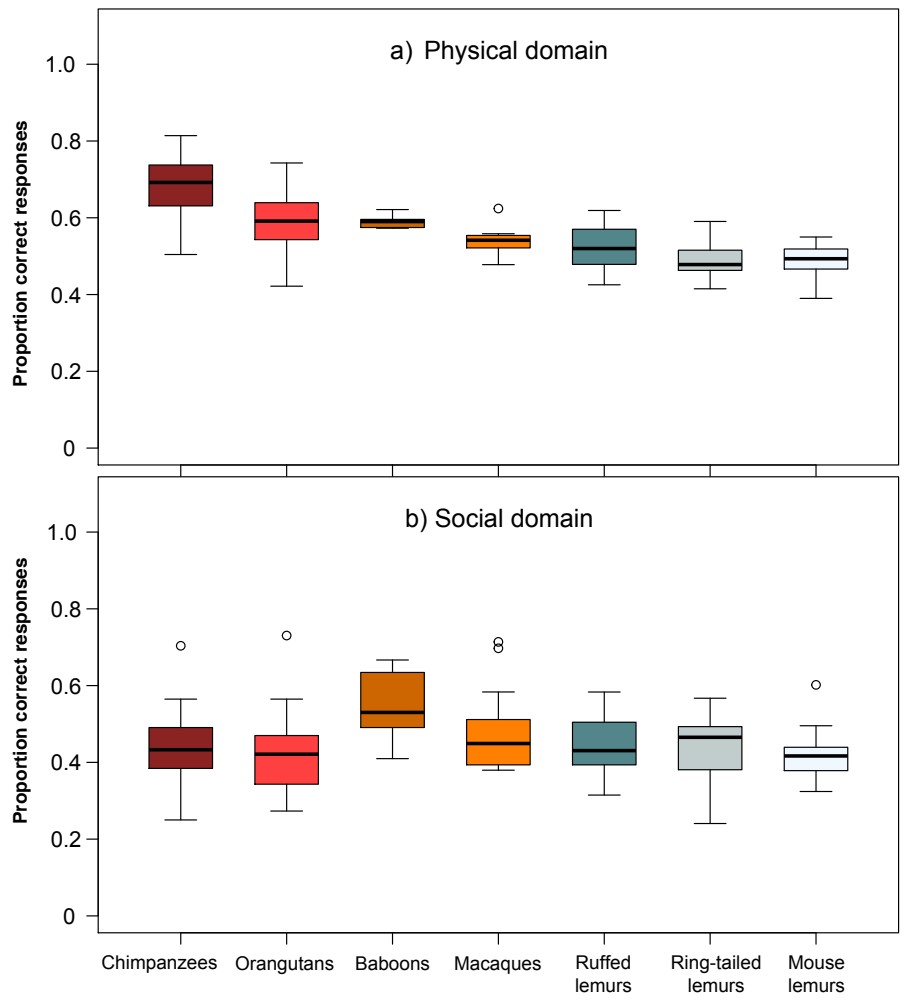

**Figure 2 Performance of the seven primate species in the (A) physical and (B) social domain.** Average performance of apes & monkeys (red and orange colours) and lemurs (blue and grey colours) in the two cognitive domains. Represented are medians (black bars), interquartile ranges (boxes), and outliers (circles).

## DISCUSSION

In this study, we applied the Primate Cognition Test Battery to three lemur species differing in socioecological traits and brain size and compared their performance with that of four haplorhine species tested in previous studies with the exact same methods. In the *physical domain*, apes and baboons performed better than lemurs in the *space scale*, chimpanzees performed better than ring-tailed lemurs in the *quantities scale* and better than mouse lemurs in the *causality scale*, after excluding the tool use task. In the *social domain*, lemurs performed at level to apes and monkeys. Most interestingly, in the *Theory of Mind scale*, great apes were outperformed by all other species except mouse lemurs. Since these species differ in relative and absolute brains size (Table 1), with a more than 200-fold difference in brain size between mouse lemurs and orangutans or chimpanzees, our results do not
**Table 3 Species differences in the six cognitive scales.** Univariate analyses for the species differences for the six cognitive scales.

| ANOVAs | Df | *F*-value | *P*-value |
|---|---|---|---|
| Quantity | 6 | 3.49 | **0.0026**[**] |
| Communication | 6 | 2.10 | 0.0549 |
| **Kruskal–Wallis tests** | **Df** | $\chi^2$ | **P-value** |
| Space | 6 | 111.68 | **<0.001**[***] |
| Causality | 6 | 68.59 | **<0.001**[***] |
| Social learning | 6 | 20.17 | **0.0026**[**] |
| Theory of mind | 6 | 55.08 | **<0.001**[***] |

**Notes.**
[**] < 0.01.
[***] <0.001 - significance levels.
Numbers in boldface: Significant deviations from chance level (Wilcoxon tests).

support the notion of a clear-cut link between brain size and cognitive skills, but suggest a more domain-specific distribution of cognitive abilities in primates.

In the *physical domain*, lemurs were outperformed by apes and baboons in the *space scale*. The species with the largest brains (apes and baboons) performed better than all other species, supporting the *General intelligence hypothesis*. These findings are in line with an earlier study showing that apes and monkeys differ in their ability to track object displacements (*Amici, Aureli & Call, 2010*). Spatial understanding is also important to remember food resources or to track conspecifics (*Dunbar & Shultz, 2017*), and species (chimpanzees, orangutans, baboons) having a larger dietary breadth performed better in these tasks, but the species with the highest amount of fruits in the diet (ruffed lemurs) did not perform better than other species, providing only partial support for the *Ecological intelligence hypothesis*. There was no clear pattern between group size and performance in the *space scale*, providing no support for the *Social intelligence hypothesis*.

In the *quantities scale*, only chimpanzees performed better than ring-tailed lemurs, and all other species performed similarly, indicating that a certain level of numerical understanding appears to be a basal cognitive trait of all primates. These results support earlier studies indicating that lemurs do not differ from haplorhine primates in numerosities and simple arithmetic operations (*Jones & Brannon, 2012*; *Merritt et al., 2011*; *Santos, Barnes & Mahajan, 2005a*). Since a comparable numerical understanding as tested in the PCTB has also been reported for various taxa outside the primate order, including fish and insects (e.g., *Agrillo et al., 2012*; *Chittka & Geiger, 1995*; *Pahl, Si & Zhang, 2013*; but see *Krasheninnikova et al., 2019*), a basal numerical understanding may be present in many animals.

In the *causality scale*, lemurs performed as well as both monkey species, but all monkeys and lemurs were outperformed by chimpanzees, who excelled in the tool use task. Even natural tool users, such as orangutans and long-tailed macaques (*Brotcorne et al., 2017*; *Van Schaik, Fox & Fechtman, 2003*), hardly solved this task (*Schmitt, Pankau & Fischer, 2012*). It required the ability to use a stick to rake a food reward into reach, which might have been too challenging for species exhibiting either a medium (baboons, macaques) or
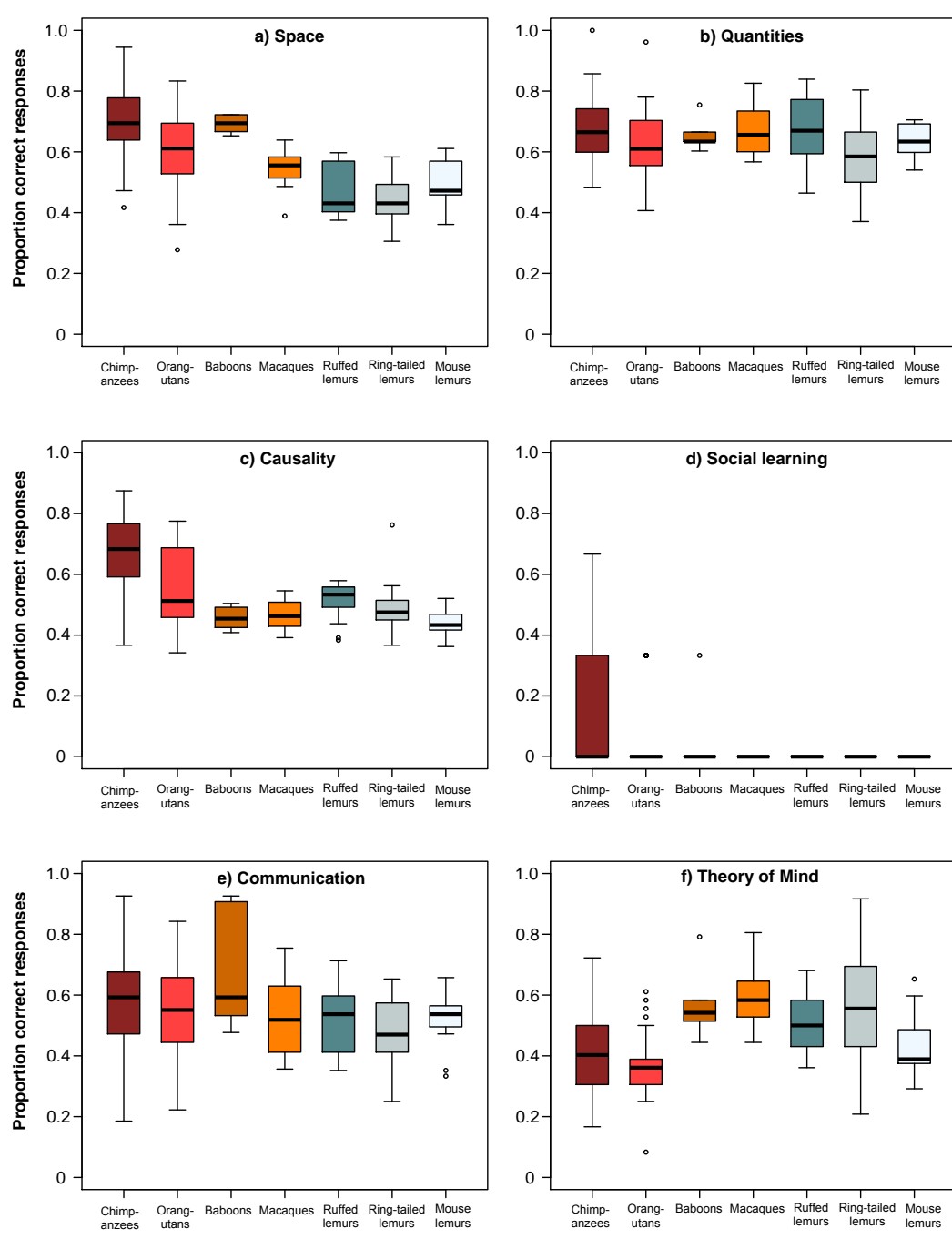

**Figure 3  Performance of the seven primate species in the six cognitive scales.** Average performance of apes & monkeys (red and orange colours) and lemurs (blue and grey colours) over the six scales. (A) Space, (B) Quantities, (C) Causality, (D) Social learning, (E) Communication, and (F) Theory of mind. Represented are medians (black bars), interquartile ranges (boxes), and outliers (circles).

**Table 4  Comparisons of performance among the seven non-human primate species for all six scales of the PCTB. Presented are the results of** *post hoc* **multiple comparisons (Bonferroni); significant results are in boldface.**  Causality II: The scale causality without the tools use task.

| | Space | Quantity | Causality | Causality II | Social learning | Communication | Theory of mind |
|---|---|---|---|---|---|---|---|
| Chimp - Orang | **<0.001** | 0.275 | **<0.001** | 1 | 1 | 1 | 1 |
| Chimp - Baboon | 1 | 1 | **0.003** | 1 | 1 | 1 | 0.082 |
| Chimp - Macaque | **<0.001** | 1 | **<0.001** | 1 | 0.699 | 1 | **<0.001** |
| Chimp - Ruffed lemur | **<0.001** | 1 | **<0.001** | 1 | 0.352 | 1 | 0.077 |
| Chimp - Ring-tailed lemur | **<0.001** | **<0.001** | **<0.001** | 1 | **0.025** | 0.29 | **<0.001** |
| Chimp - Mouse lemur | **<0.001** | 1 | **<0.001** | **0.041** | 0.229 | 1 | 1 |
| Orang - Baboon | 1 | 1 | 1 | 1 | 1 | 0.677 | **0.014** |
| Orang - Macaque | 1 | 1 | 0.433 | 1 | 1 | 1 | **<0.001** |
| Orang - Ruffed lemur | **0.004** | 1 | 1 | 0.560 | 1 | 1 | **0.009** |
| Orang - Ring-tailed lemur | **<0.001** | 1 | 0.643 | 1 | 0.919 | 1 | **<0.001** |
| Orang - Mouse lemur | 0.237 | 1 | **0.046** | 0.918 | 1 | 1 | 1 |
| Baboon - Macaque | 0.176 | 1 | 1 | 1 | 1 | 0.591 | 1 |
| Baboon - Ruffed lemur | **0.001** | 1 | 1 | 1 | 1 | 0.653 | 1 |
| Baboon - Ring-tailed lemur | **<0.001** | 1 | 1 | 1 | 1 | 0.094 | 1 |
| Baboon - Mouse lemur | **0.023** | 1 | 1 | 1 | 1 | 0.424 | 0.816 |
| Macaque - Ruffed lemur | 1 | 1 | 1 | 1 | 1 | 1 | 1 |
| Macaque - Ring-tailed lemur | 0.074 | 0.307 | 1 | 1 | 1 | 1 | 1 |
| Macaque - Mouse lemur | 1 | 1 | 1 | 1 | 1 | 1 | **0.033** |
| Ruffed lemur - Ring-tailed lemur | 1 | 0.409 | 1 | 1 | 1 | 1 | 1 |
| Ruffed lemur - Mouse lemur | 1 | 1 | 1 | **0.008** | 1 | 1 | 1 |
| Ring-tailed lemur - Mouse lemur | 1 | 1 | 1 | 0.106 | 1 | 1 | **0.036** |

low (lemurs) level of precision grip (*Torigoe, 1985*). Although long-tailed macaques use stone tools to crack open nuts or mussels, they do so mainly by applying force rather than using fine-motor skills (*Gumert & Malaivijitnond, 2012*). Thus, the tool use task appears unsuitable for a fair interspecific comparison. Excluding this task from the *causality scale* resulted in a rather similar overall average performance of all species. Interestingly, lemurs that have never been observed to use tools in the wild (*Fichtel & Kappeler, 2010*; *Kittler, Schnoell & Fichtel, 2015*; *Kittler, Kappeler & Fichtel, 2018*), appeared to exhibit an understanding for the necessary functional properties of pulling tools (*Santos, Mahajan & Barnes, 2005b*; *Kittler, Kappeler & Fichtel, 2018*). Hence, except for the *space scale* we did not find systematic species differences in performance, challenging the notion that there is a domain-general distinction between haplorhines and strepsirrhines (*Deaner, Van Schaik & Johnson, 2006*). Our results instead suggest the existence of domain-specific cognitive differences.

In the social domain, species differences were less pronounced, and lemurs' overall performance in the *Theory of Mind scale* was equal to that of monkeys and even superior to that of apes. In the *social learning scale* neither lemurs, nor baboons or long-tailed macaques solved the task. However, long-tailed macaques exhibit cultural variation in stone handling techniques in the wild, indicating that they are able to learn socially (*Brotcorne et al.,*

*2017*). The ability to learn socially has also been reported in ring-tailed and ruffed lemurs (e.g., *Kappeler, 1987*; *Kendal et al., 2010*; *O'Mara & Hickey, 2012*; *Stoinski, Drayton & Price, 2011*), but remains unstudied in mouse lemurs. Since individuals had to learn in this task from a human demonstrator, the phylogenetic distance between species and the demonstrator might have influenced learning abilities, because great apes performed better than Old World monkeys and lemurs (*Schmitt, Pankau & Fischer, 2012*). Hence, it remains an open question whether monkeys and lemurs would perform better when tested with a conspecific demonstrator. Moreover, the task required the ability to shake a transparent tube or to insert a stick into the tube, which might have been too difficult for species with limited dexterity (*Torigoe, 1985*). Therefore, a social learning task demonstrated by a conspecific and adapted to manipulative skills of Old World monkeys and lemurs (*Schnoell & Fichtel, 2012*), posing technical problems that they have to face in their natural environment (*Kummer & Goodall, 1985*), might be more informative in future studies

In the *communication scale*, all species performed equally well, suggesting that all species can make use of socio-visual cues given by others. This result is in line with those of several other studies showing the ability to use social-visual cues presented by a human demonstrator in object-choice experiments in birds (*Schmitt, Pankau & Fischer, 2012*), aquatic mammals (sea lions: *Malassis & Delfour, 2015*; dolphins: *Tschudin et al., 2001*), domestic animals (dogs: *Kaminski et al., 2005*; *Miklósi et al., 1998*); pigs: *Nawroth, Ebersbach & von Borell, 2016*; goats: *Wallis et al., 2015*), as well as other primates (*Anderson & Mitchell, 1999*; *Itakura, 1996*).

In contrast, unexpected species differences emerged in the *Theory of Mind scale,* with great apes performing inferior to both monkeys and lemurs. This difference was mainly due to better performance of monkeys and lemurs in the intentions task, but not in the gaze following task. In the gaze following task, all lemurs performed below chance level, although it has been shown that ring-tailed lemurs follow the gaze of conspecifics (*Shepherd & Platt, 2008*) and that they use human head orientation as a cue for gaze orientation in a food choice paradigm (*Botting, Wiper & Anderson, 2011*; *Sandel, MacLean & Hare, 2011*), questioning the validity of these gaze following tasks. In the intention task, a human observer tried to reach a cup with a hidden reward repeatedly with the hand. Monkeys and lemurs might have performed better than apes because they may have solved the task by using spatial associations between the repeated hand movements and the cup or by understanding the hand movements as a local enhancement (*Shettleworth, 2010*; *Schmitt, Pankau & Fischer, 2012*). Still, it remains puzzling why chimpanzees and orangutans did not use the hand movement as a cue for the location of the hidden reward. Even more so because a comparative study of *Theory of Mind* compatible learning styles in a simple dyadic game between seven primate species, including chimpanzees and ring-tailed lemurs, and a competitive human experimenter revealed that test performance was positively correlated with brain volume, but not with social group size, suggesting that *Theory of Mind* is mostly determined by general cognitive capacity (*Devaine et al., 2017*). Hence, additional social cognitive tests are required to obtain a better understanding of the relationship between brain size and cognitive performance in the social domain.

Altogether, average species performances were generally not as different as it might have been expected in view of the various hypotheses on the evolution of cognitive abilities. Except for the *space scale*, the overall comparison did not provide support for the *General intelligence hypothesis*, since variation in brain size cannot explain the observed results. Similarly, performances of the seven species did not reflect any clear patterns concerning their feeding ecology, i.e., the percentage of fruit in the diet or dietary breadth, except for the *space scale* (see Table 1); hence, these results did not provide support for the *Ecological intelligence hypothesis*. Moreover, our results did not provide support for the *Social intelligence hypothesis* because lemurs, and especially the solitary mouse lemurs, should have performed inferior compared to the haplorhine species (*Dunbar & Shultz, 2017*).

Earlier comparative studies among primates linking performance in a range of comparable cognitive tests in the physical or social domain revealed a link between performance in these tasks and brain size (*Deaner, Van Schaik & Johnson, 2006*; *Deaner et al., 2007*; *Reader & Lal, 2002*; *Reader, Hager & Lal, 2011*). However, studies using the exact same experimental set up revealed contradictory results. Two studies addressing only one cognitive ability revealed a positive relationship between brain size and performance in inhibitory control or *Theory of Mind* tests (*MacLean et al., 2014*; *Devaine et al., 2017*), but all other studies applying various tests on inhibitory control and spatial memory (*Amici, Aureli & Call, 2008*; *Amici, Aureli & Call, 2010*; *Amici et al., 2012*) or tasks of the Primate Cognition Test Battery (*Herrmann et al., 2007*; *Schmitt, Pankau & Fischer, 2012*); this study), found no clear-cut relationship between brain size and cognitive performance.

Even though lemurs performed at level with monkeys and great apes in many of these experiments, we do not suggest that their cognitive abilities are *per se* on par with those of larger-brained primates. In the physical domain, species differences emerged only in the *space scale*, with species having larger brains performing better. These findings might provide support for the *General intelligence hypothesis* but the sample size is rather small to make any firm conclusions. However, no systematic species differences were found in the *quantities* or *causality scales*, which might not be variable enough to reveal actual differences between species. Since some fish and insects possess similar basal cognitive skills as tested in the physical domain of the PCTB (*Fuss, Bleckmann & Schluessel, 2014*; *Loukola et al., 2017*; *Schluessel, Herzog & Scherpenstein, 2015*), the potential link between brain size and performance in the *space scale* requires further testing. In the social domain, the social learning task was not suitable for all species, and individuals might have recruited other abilities to solve the problems, as discussed for the intention task above. Hence, to examine species differences in cognitive abilities, it is necessary to conduct cognitive tests that measure the cognitive abilities they are intended to measure and that are sensitive, i.e., difficult enough to detect variation in cognitive performance without producing ceiling or floor effects (see also *Schubiger, Fichtel & Burkart, 2020*).

In addition, many tests of the PCTB were based on two-or three-choice paradigms in which the costs for choosing correctly were rather low, because the probability to receive a reward was either 50% or 33%, and a random choice strategy might have been still relatively profitable. For example, performance in a memory task increased in common

marmosets (*Callithrix jacchus*) and common squirrel monkeys (*Saimiri sciureus*) from a two-choice task to a nine-choice task, in which the probability of success was lowered from 50% to 11%, making a wrong choice more costly, appeared to favour an appropriate learning strategy over a random choice strategy (*Schubiger, Kissling & Burkart, 2016*). The application of a random choice strategy may also explain why four parrot species that were tested with the PCTB, may have failed to solve the tasks, besides morphological differences in performing the tasks (*Krasheninnikova et al., 2019*).

Finally, the PCTB was designed to examine the spontaneous ability to solve the tasks, and not to examine how long individuals need to learn the task. Hence, a test battery that continued testing until individuals reached a certain criterion (e.g., 80% correct responses) or detailed analyses of applied learning strategies as in *Devaine et al. (2017)* may allow to compare not only species differences in their spontaneous ability to solve the task, but also species-specific learning curves as well as learning strategies, which might reveal more informative differences.

## CONCLUSIONS

To conclude, our study generated the first systematic results on cognitive abilities in lemurs, and the comparison with haplorhines suggests that in many aspects of the physical and social domain, the average performance in these tests of members of these two lineages do not differ substantially from each other. These results, which are based on a small sample size, reject the notion of a direct correlation between brain size and cognitive abilities assessed in the PCTB and, may question assumptions of domain-general cognitive skills in primates. Overall, our results strengthen the view that when comparing cognitive abilities among species, it is of vital importance to include a diverse set of tests from both cognitive domains that are applicable to a diverse range of species and taxa (*Auersperg et al., 2011*; *Auersperg, Gajdon & von Bayern, 2013*; *Burkart, Schubiger & Van Schaik, 2016*; *Maclean et al., 2012*; *Schmitt, Pankau & Fischer, 2012*) and to carefully consider the internal and external validity of the specific tests (*Krasheninnikova et al., 2019*; *Schubiger, Fichtel & Burkart, 2020*).

## ACKNOWLEDGEMENTS

We are grateful to Silvio Dietzel and the "Erlebnispark Affenwald" for permission to work with the lemurs. We would also like to thank Esther Herrmann and Vanessa Schmitt for sharing the PCTB performance data of the great apes and monkeys with us. Furthermore, we are grateful to Ulrike Walbaum, Anna Zango Palau, Luise Zieba and Lluis Socias Martinez for helping with the experiments and inter-observer coding the videos. Thanks to Sarah Hartung, Henry Benseler and Ramona Lenzner-Pollmann for taking care of the animals. We are also grateful to Andrew Zamora, Louise Barrett and an anonymous reviewer for very constructive comments on an earlier version of the manuscript.

### Funding

This study was supported by the German Science Foundation (Deutsche Forschungsgemeinschaft; awarded to Claudia Fichtel: FI 929/8-1). The funders had no role in study design, data collection and analysis, decision to publish, or preparation of the manuscript.

### Grant Disclosures

The following grant information was disclosed by the authors:
German Science Foundation: FI 929/8-1.

### Competing Interests

The authors declare there are no competing interests.

### Author Contributions

- Claudia Fichtel conceived and designed the experiments, analyzed the data, prepared figures and/or tables, authored or reviewed drafts of the paper, and approved the final draft.
- Klara Dinter conceived and designed the experiments, performed the experiments, analyzed the data, prepared figures and/or tables, authored or reviewed drafts of the paper, and approved the final draft.
- Peter Kappeler conceived and designed the experiments, authored or reviewed drafts of the paper, and approved the final draft.

### Animal Ethics

The following information was supplied relating to ethical approvals (i.e., approving body and any reference numbers):

The Animal Welfare Body of the German Primate Center approved the experimental protocol (E2-17).

### Data Availability

The raw performance data of lemurs in the Primate Cognition Test Battery are available in the Supplementary Files. Performance data of the four haplorhine primate species were previously published in *Herrmann et al. (2007)* and *Schmitt, Pankau & Fischer (2012)*.

### Supplemental Information

Supplemental information for this article can be found online at http://dx.doi.org/10.7717/peerj.10025#supplemental-information.

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
