# Peer review of "The lemur baseline: how lemurs compare to monkeys and apes in the Primate Cognition Test Battery"

_PeerJ, doi:10.7717/peerj.10025_

## Round 0.1 · original submission · Minor Revisions

Dear Claudia,

Thank you for your submission, which has now been seen by two reviewers. Both view your paper positively and consider it will make a valuable contribution to the literature; an assessment with which I agree enthusiastically. The reviewers also have some suggestions for improvement, and again I agree that these changes will further improve your paper. I would be very grateful if you could give these comments your close attention. I look forward to seeing a new version soon.

with best wishes,

Lou

Reviewer 1 ·

Basic reporting

The manuscript is well written, the raw data are supplied, and the figures are relevant and serve to illustrate the data.
The PCBT has been replicated with chimpanzees (Hopkins et al., https://doi.org/10.1016/j.cub.2014.05.076 ). Given that only few studies used this test battery, I think this study should be mentioned.
Line 249: unclear what “control task” refers to in this context.
Line 322/323: the statement that there were no significant differences in the causality scale after excluding the tool use task does not appear to match table 4 (the Chimp – Mouse Lemur comparison). If the authors refer here to table 3: this table does not appear to include the causality scale without the tool use task.
Line 356: “causality scale” not “quantity scale”.
Table 2: the table does not show the cell with the “mouse lemur” label. The reporting of the 95% CI is confusing. There should be a lower and an upper value. Currently the impression is that there is only one value with two decimals.
Figure 1 - 3: how are outliers defined? Not sure what the authors mean by “hinges (whiskers)”. I think the ends of the box (defined as the interquartile range) should indicate the hinges (25th Percentile, 75th Percentile).
The dashed lines in the figures should be defined as well (chance level?). Why are the dashed lines shown in figure 1 but not in figure 3 (at least for the facets in which all tasks have the same chance value)? For the tool use task (Fig 1), there should be no chance level.
In which order were the tasks conducted?
Figure 1: Communication: Pointing: Ruffed lemurs: the outlier point appears to be shifted to left.
Figure 3: title of the lower right facet it missing.
Data file: the authors use commas as decimal separator and semicolons as column separator. I think this should be changed to a decimal point and comma as column separator. Additionally, a legend would be helpful (e.g., which conditions are included for each task). It would be valuable if the data could be provided in a more detailed format (i.e., including the conditions within each task).
It would be helpful if more details regarding test experience of the subjects could be added. Were they trained in pointing / object choice tasks before?
Stating the initial sample size (including any dropouts) in addition to the final (analysed) sample size would be useful. If there were no dropouts, it would be good to mention this in the methods section.

Experimental design

Line 188: It would useful to add the interobserver reliability for every task (e.g., in a table in the supplementary material).
Line 246, Supplementary material: Line 35 /36: “Individuals had to choose both cups correctly to count as a correct response.” Doesn’t that mean that the chance level here should be 2/3 * 1/3 = 2/9 and not 1/3 as indicated in Fig 1 and Table 2?
Supplementary material: line 56/57: “Individuals had to choose the reward-cup as first choice to count as a correct response.” Does “reward-cup” refer to the cup where the reward ends up or to one of the two cups where the reward could have ended up (in double displacements)? If it is the cup, where the reward was actually located the maximal performance should be 2/3 and not 1 in the double displacement trials, correct? This might be relevant for the interpretation of the performance in this task.
Supplementary material: line 113/114: if an individual had ignored the quantities presented on the middle plates, they would have scored 10/14 trials correct, right? If that is the case, I wonder whether we can be certain that this task measures addition (rather than relative quantities). It would be interesting to examine how the subjects performed in the trials in which the addition resulted in a reversal of the location of the larger amount of food.
Line 258 and Supplementary material 247/247: Figure 1 gives the impression that there would be a chance level for the task “attentional state” but that does not seem to be the case. In Table 2, the authors do not specify a chance value for “attentional state” so it is unclear what the statement in line 258 refers to.

Validity of the findings

Line 232: details concerning assumption checks for the analyses should be reported somewhere. The authors note (line 234) that they used parametric and non-parametric analyses depending on the tasks. It would be helpful to mention how they made these decisions. Were the assumptions for the MANOVAs met (absence of multicollinearity for the dependent variables; are dependent variables multivariate normally distributed within each group of the independent variables?)?
Line 428-430: one could have made exactly the same argument about the spatial scale if the results had been the other way round (i.e. spatial cognition could be seen as “rather basal cognitive abilities” compared to causal / numerical cognition).
The authors refer to external validity towards the end of their manuscript. What about internal validity of the scales used to assess cognitive abilities? The authors correlate their personality and inhibitory control measures with their physical and social cognition tasks but they do not report correlations for their different physical and social cognition scales. Though this might not be the central focus of interest in this study it would be valuable examine the construct validity of the scales (e.g. to what extent are the conditions within the tool property scale correlated?). Previous research found only limited evidence for contextual repeatability / convergent validity in comparative cognitive research (e.g., Herrmann and Call, Phil. Trans. R. Soc. B, 2012; Cauchoix et al., Phil. Trans. R. Soc. B, 2018; Voelter et al., Phil. Trans. R. Soc. B, 2018).

Additional comments

In this study, the authors conducted the primate cognition test battery with three lemur species. I think this is a very valuable contribution to the literature given how few studies applied extensive test batteries in primate cognition research to data. The manuscript is clearly written in professional, unambiguous language. There are a number of questions I had regarding the analyses, particularly to what extent the assumptions of the statistical methods were met and regarding the chance level for some of the tasks. These points should be clarified upon before acceptance.

Line 134-140: One could make the point that these hypotheses are not necessarily mutually exclusive and different ecological and physiological factors might be linked to the evolution of cognitive abilities.
Line 454-456: I think the conclusion here is a bit too strong given the limited sample size of the current study (both in terms of individuals and species). It might be that with more power (a larger number of species) one would detect a correlation between cognitive abilities and brain size. Additionally, this study does not report any correlation analyses between brain size and cognitive ability.

·

Basic reporting

Line 50
Further detail on what exactly the “enhanced cognitive skills” possessed by primates are would be beneficial from the perspective of readers who might take such a statement for granted without necessarily thinking through what it means.

Lines 79-80
Similarly, some examples on how cognitive tests when used across species can fail due to various aspects of their methodology would also likely be appreciated by several readers.

Line 101
I’m a bit skeptical of Schmitt et al.’s 2011 study if only because of the combination of three factors: a small sample size 13 M. fascicularis and 5 P. anubis), the large range of variation in how individuals from both species performed in some tasks, and how performance was close to 50% across several tasks. None of this is the fault of the author’s of the current study of course, but highlights how difficult research in this area can be. In addition, looking into Herrmann et al.’s paper first describing the PCTB showed that they (understandably) relied on a large population of captive/rescued apes that presumably have been interacting with human beings extensively. Hard to see how that wouldn’t affect their cognitive behavior, and Tomasello and Call made this very point in their 2004 commentary in Animal Cognition (“The role of humans in the cognitive development of apes revisited”) retrospectively looking at their previous research. Although children have also been interacting with human being extensively as well and perhaps a balanced comparison is being made!

Line 105
More of a suggestion for a stimulating book. If the possibility of varieties in social “styles” potentially being reflected in cognition/brain development is of interested I would highly recommend Jaak Panksepp’s book Affective Neuroscience. His circuit-based view of the brain, particularly it’s emotive centers, is sure to evoke some compelling thoughts in primatologists.

Line 122
Small grammatical/lexical suggestion: change “seems indicated” to “is warranted”

Line 125
Another small lexical suggestion: change “in the following” to “hereafter”

Line 130
Do the authors mean that the average sub-group size in ruffed lemurs is six individuals? The ruffed lemur communities at all three study sites covered by the studies the authors cite have well over 10 individuals. As ruffed lemur biologists have pointed out, one can’t call these socially dispersed communities groups in the strict sense but the sentence as currently written might mislead the reader. Apologies for being nit-picky!


Lines 200-208 (with suggested edits applying hereafter)
I would place the word “scale” after rather than before the type of scale being described e.g. space scale instead of scale space, quantity scale instead of scale quantity, etc.

Table 1 legend
I would suggest adding “Summary of the most important socio-ecological traits for the seven non-human primate species considered in the present study”


Table 2
I am not sure if this is a typographic error or a function of my being an American, but it seems that periods were used to separate confidence interval values here? Personally I was confused and easily mistook the pairs of numbers as being from a single value e.g. 51.58% for ruffed lemurs performance in the Space tasks. Perhaps add more space between the interval values?

Figures
Across all multi-panel figures, one can see that tick marks and axis labels are being covered by a figure’s horizontal or vertical neighbors, resulting in lines bridging the space between neighboring plots. If these figures were made in R using ggplot2, this can easily be avoided by removing the tick marks and axis labels for the relevant plots.

Figure 3
The Theory of Mind panel in the bottom right is lacking a title

Experimental design

Line 157
Would the authors consider making some of the recordings available as supplementary files? Readers (and the public) would surely get a kick out of seeing what administering the PCTM to these lemurs looked like.

Lines 200-208 (with suggested edits applying hereafter)
I would place the word “scale” after rather than before the type of scale being described e.g. space scale instead of scale space, quantity scale instead of scale quantity, etc.

Line 227
In the future, the authors may wish to consider presenting the raw, multiple-test adjusted results alongside results from creating simulated datasets modeling responses to the PCTB tasks as completely random (e.g. from a normal distribution centered on .5 with several different standard deviations) to estimate how robust their findings are to false positive rates. This seems especially pertinent given the common pattern of PCTB scale results being close to 50%.

Line 230
Similar simulations (or boostrapping/jack-knifing procedures) as suggested above can be used to estimate how robust the MANOVA and ANCOVA results are to variations in the sample sizes/characteristics of the covariates considered.

Validity of the findings

Lines 329 – 331
Is it possible to distinguish support for the general intelligence hypothesis from the ecological intelligence hypothesis here? Chimps and baboons range over larger areas than the other species being considered and this would likely select for better performance on the tasks of the space scale. Although, one would then expect, say, ruffed lemurs and ring-tailed lemurs to outperform mouse lemurs on this scale which they didn’t.

Figure 2 and Lines 364-378 (also relevant for Lines 435-444)
I’m grateful for the opportunity that the authors have presented readers like myself to see result for the same cognitive test applied across a wide range of species. One aspect that bears highlighting is just how surprised I was at the near-uniform performance of primates across the Social domain. Especially because they all hover quite closely to a 50% proportion correct. A few questions and comments come to mind which could also be applied to the Physical domain of cognitive tasks:

1. Is this average value a meaningful quantity given that it summarizes scores on different tasks, that presumably have different underlying neurological mechanisms?
2. One is tempted to see 50% as “no better than a coin flip” but that raises the question of what differences in performing better or worse on these tasks mean ecologically. What does scoring 60% along the social domain vs 40% translate to in the life of these primates? What are the potential benefits/costs?
3. Could making the correct choices in these kinds of tasks 50% of the time actually be “good enough” for whatever the effects would be within an individual’s ecological environment?

Lines 386-404
Many have noted (e.g. Kummer and Goodall’s 1985 paper in the Philosophical Transactions of the Royal Society of London “Conditions of innovative behavior in primates”) that replicating the ecological and motivational conditions under which primates would naturally be acting is likely key to really understanding their cognitive abilities. The tool use and social learning tasks in the PCTB, for example, are very much unlike any situation that the species studied would find themselves in. Brian Hare has plenty of published work demonstrating that chimpanzees, for example, perform better in theory of mind tasks when placed in a competitive environment.

Lines 405-414
Variation in brain size may not map onto variation in other functional aspects of the central nervous system. For example, it’s potentially telling that the only axis on the PCTB that chimpanzees dramatically outperformed macaques and baboons on in Schmitt et al.’s (2011) study were on tool use and one of the theory of mind tasks – both of which are cognitive dimensions heavily affected by motivation. The authors of the present study found similar results.

In their 2018 PNAS paper “A neurochemical hypothesis for the origin of hominids”, Raghanti and her colleagues found that chimpanzees and humans had elevated dopamine levels in two brain areas compared to Cebus, Macaca, Papio, and Gorilla individuals. Dopamine plays a substantial role in regulating motivational processes! I once more encourage the authors to read Affective Neuroscience.

I would also note that in the broad sense intelligence, whether general, ecological, social, etc., is an emergent property that I would be skeptical of being described well by the performance on individual tasks in the PCTB.

Line 455
The authors did a commendable job of viewing their results with humility up to this point. I would urge them to continue exercising that humility before stating in a factual sense that a direct correlation between brain size and “cognitive abilities” has been rejected by this study, given that whether the PCTB really measures cognitive abilities is, in my own opinion, an open question.

Additional comments

It was a pleasure reading a manuscript covering similar topics to my undergraduate honors thesis. I confess that my interests in primate cognition fall very short of expertise, but do hope that my forays into that realm combined with a more rigorous background in behavioral ecology yielded useful comments below.

The authors deserve applause for putting yet another cognitive battery to the test, especially for expanding it into Strepsirrhines. The manuscript is well-written and detailed, the methodology is sound in so far as a previously developed test is being used in a standardized way, and the results come with some potential surprises that will compel others in the field to dig deeper into Strepsirrhine cognition and also meditate on the measure of primate intelligence overall.

The authors are also humble in the discussion of their results and what the implications therein may be. It is here however, that the manuscript could benefit from some additional detail. While I know that we are wary of “hand-waving”, the authors are familiar enough with lemur behavior, ecology, and evolution, to give us a little more speculation as to why they found what they found. Especially in light of the general lack of support for the “general intelligence hypothesis”, many readers would appreciate the opportunity to hear more about why lemurs may exhibit their own kind of intelligence that is particularly well-suited for the environment and constellation of traits (if I may, I am a fan of the Lemur Syndrome) that make them stand apart from other primates. Much of what I’ve just said is, however, personal preference and I would leave it up to the editor and authors to decide of the readership of PeerJ would benefit from the addition of these details.

Warm Regards,
Andrew J Zamora

---

## Round 0.2 · accepted · Accept

Thanks for your attention to these suggestions, Claudia, and I'm delighted to accept your paper for publication!